# Adversarial Projections to Tackle Support-Query Shifts in Few-Shot Meta-Learning

**Aroof Aimen**[*1]  **Bharat Ladrecha**[*1]  **Narayanan C. Krishnan**[2]

[1]Indian Institute of Technology, Ropar
[2]Indian Institute of Technology, Palakkad

**Abstract**  Popular few-shot Meta-learning (ML) methods presume that a task's support and query data are drawn from a common distribution. A recent work relaxed this assumption to propose a few-shot setting where the support and query distributions differ, with disjoint yet related meta-train and meta-test support-query shifts (SQS). We relax this assumption further to a more pragmatic SQS setting (SQS+) where the meta-test SQS is unknown and need not be related to the meta-train SQS. The state-of-the-art solution to address SQS is transductive, requiring unlabelled meta-test query data to bridge the support and query distribution gap. In contrast, we propose a theoretically grounded inductive solution - Adversarial Query Projection (AQP) for addressing SQS+ and SQS. AQP can be easily integrated into the popular ML frameworks. Exhaustive empirical investigations on benchmark datasets and their extensions, different ML approaches, and architectures establish AQP's efficacy in handling SQS+ and SQS.

## 1 Introduction

Meta-learning (ML) approaches assume that the meta-train and meta-test tasks are drawn from a common distribution. The shared distribution assumption prevents the use of meta-learned models in evolving test environments deviating from the training set. Recent ML works attempt at relaxing this assumption [15, 13]. However, these ML approaches assume a common distribution inside the tasks, i.e., the task-train and task-test data come from the same distribution. But a distribution shift may exist between the task-train data (support set) and task-test data (query set) because of the evolving or deteriorating nature of real-world objects or environments, differences in the data acquisition techniques from support to query sets, extreme data deficiency from one distribution, etc. Addressing support query shift (SQS) inside a task has gained attention very recently [3]. However, this pioneering work assumes the prior knowledge of SQS in the meta-test set and induces a related although disjoint SQS in the meta-train set. The model trained on such a meta-train set is accustomed to handle the SQS and, to some extent, becomes robust to the related unseen meta-test SQS. In this paper, we consider, SQS+, a more generic SQS problem where the prior knowledge of the meta-test SQS is absent. We expect an unknown SQS in the meta-test set and therefore cannot induce any related SQS in the meta-train set. The earlier work on addressing SQS [3] is a limiting case of SQS+. The solution to SQS proposed by Bennequin et al., [3] uses optimal transport (OT) to bridge the gap between support and query distributions, but assumes the availability of unlabelled query during testing. While this solution can be adopted for our proposed problem, access to unlabelled query data during meta-test may be unrealistic in many real-world scenarios. Our solution to address the support query (SQ) shift problem - Adversarial Query Projection (AQP), does not require transduction during meta-testing and thus is applicable in such real-world scenarios.

Overall, we make the following contributions:

- We propose, SQS+, a practical SQS setting for few-shot meta-learning. The shift between support and query sets during meta-testing is unknown while meta-training the model.

- We contribute to the FewShiftBed [3] realistic datasets for evaluating methods that address SQS and SQS+. In these datasets, meta-train data lacks SQS while meta-test data contains SQS.

- We design an inductive solution for tackling SQS+ using adversarial query projections (AQP). The AQP module is standalone and could be integrated with any few-shot ML episodic training regimen. We verify this capability by integrating AQP into Prototypical (ProtoNet) and Matching Networks (MatchingNet).

- Exhaustive empirical investigation validates the effectiveness of the AQP on various settings and datasets, preventing a negative impact even in the absence of SQS.

## 2 Related Work

Transductive meta-learning approaches that utilize unlabeled query data in the training process are effective baselines for handling SQS in few-shot learning (FSL). Ren et al., [12] introduce a transductive prototypical network that refines the learned prototypes with cluster assignments of unlabelled query examples. Boudiaf et al. [4] induce transduction by maximizing the mutual information between query features and their predicted labels in conjunction with minimizing cross-entropy loss on the support set. Minimizing the entropy of the unlabeled query instance predictions during adaptation [5] also achieves a similar goal. Liu et al., [10] propose a graph based label propagation from the support to the unlabeled query set that exploits the data manifold properties to improve the efficiency of adaptation . Antoniou et al., [1] show that minimizing a parameterized label-free loss function that utilizes unlabelled query data during training can also bridge SQS. Inspired from learning invariant representations [7, 2, 6], Bennequin et al. [3] use Optimal Transport (OT) [11] during meta-training and meta-testing to address SQS. In contrast, we propose an inductive method to tackle SQS in FSL where access to the unlabelled meta-test query instances is not required. Inductive approaches to tackle train-test domain shifts have relied on adversarial methods for data/task augmentations. Goldblum et al., [8] propose adversarial data augmentation for FSL setup and demonstrate the robustness of the model trained on augmented tasks to adversarial attacks at meta-test time. Wang et al. [15] bridge the shift between meta-train and meta-test domains by adversarial augmentation by constructing virtual tasks learned through adversarial perturbations. A model trained on such virtual tasks becomes resilient to meta-train and meta-test domain shifts. While adversarial perturbations are central to our approach, we use it to tackle a different problem, support query distribution shifts inside a task for FSL.

## 3 Methodology

### 3.1 Preliminaries

#### 3.1.1 Notations.
A typical ML setup has three phases - meta-train $M$, meta-validation $M_v$ and meta-test $M_t$. A model is trained on $M$ and evaluated on $M_t$. $M_v$ is used for hyperparameter tuning and model selection. The dataset $(C, \mathcal{D})$ comprising of classes and domains is partitioned into $(C_M, \mathcal{D}_M)$, $(C_{M_v}, \mathcal{D}_{M_v})$, and $(C_{M_t}, \mathcal{D}_{M_t})$ corresponding to the phases $M$, $M_v$ and $M_t$, respectively. Each phase is a collection of tasks and every task $T_0$ is composed of a support set $T_{S_0}$ and a query set $T_{Q_0}$. The support set $T_{S_0}$ and query set $T_{Q_0}$ contain (example $x$, label $y$) pairs from $N$-classes with $K$ and $Q$ examples per class, with the label of meta-test query instances being used only for evaluation.

#### 3.1.2 Support-Query Distribution Shift.
In a classical few-shot learning setup, the domain is constant across $M, M_v, M_t$ phases and within the tasks. So, in addition to a common distribution $\mathcal{T}_0$ over tasks, a shared distribution exists even at the task composition level, i.e., $\mathcal{T}_{S_0} = \mathcal{T}_{Q_0}$, where $\mathcal{T}_{S_0}$ and $\mathcal{T}_{Q_0}$ are the distributions on support and query sets respectively. A more pragmatic case is that of SQS, wherein a distribution mismatch occurs between the support and query sets within a task. Let $\mathcal{D}_M$ and $\mathcal{D}_{M_t}$ be the set of domains for the $M$ and $M_t$ phases. We skip $M_v$ for convenience, but

it follows the same characteristics as $M$ and $M_t$. We define our version of the support query shift problem termed SQS+ as follows.

**Definition 1.** *(SQS+) The support and query sets of every meta-train task come from the domain $\mathcal{D}_M$ and share a common distribution $\mathcal{T}_{S_0} = \mathcal{T}_{Q_0}$. Let $D_S^{M_t}, D_Q^{M_t} \in \mathcal{D}_{M_t}$ be the support and query domains for a meta-test task. The SQS+ setting is characterized by an unknown shift in the support and query domains of a meta-test task, $D_S^{M_t} \neq D_Q^{M_t}$ (introducing a shift in the support and query distributions $\mathcal{T}_{S_0} \neq \mathcal{T}_{Q_0}$), along with the standard SQS assumption of disjoint meta-train and meta-test domains - $\mathcal{D}_M \cap \mathcal{D}_{M_t} = \emptyset$.*

## 3.2 Adversarial Query Projection (AQP)

Without leveraging unlabelled meta-test query instances, our solution induces the hardest distribution shift for the meta-model's current state. For a task $T_0$, we simulate the worst distribution shift by adversarially perturbing its query set $T_{Q_0}$ such that the model's query loss $L^*$ maximizes. Let $H$ be the task composition space, i.e., $H$ is the distribution of support and query distributions such that $\mathcal{T}_{Q_0} \sim H$ and $\mathcal{T}_Q \sim H$. Let $T_{Q_0}$ and $T_Q$ be the samples belonging to $\mathcal{T}_{Q_0}$ and $\mathcal{T}_Q$ respectively (we occasionally denote $T_Q \sim H$ because $T_Q \sim \mathcal{T}_Q \sim H$, to improve readability). Also, let $\Theta$ be the parameter space with $\theta, \phi \sim \Theta$, and $d : H \times H \to R_+$ be the distance metric that satisfies $d(T_{Q_0}, T_{Q_0}) = 0$ and $d(T_Q, T_{Q_0}) \geq 0$. We consider a Wasserstein ball $B$ centered at $\mathcal{T}_{Q_0}$ with radius $\rho$ denoted by $B_\rho(\mathcal{T}_{Q_0})$ such that:

$$B_\rho(\mathcal{T}_{Q_0}) = \{\mathcal{T}_Q \in H : W_d(\mathcal{T}_Q, \mathcal{T}_{Q_0}) \leq \rho\}$$

where $W_d(\mathcal{T}_Q, \mathcal{T}_{Q_0}) = \inf\limits_{M \in \pi(\mathcal{T}_Q, \mathcal{T}_{Q_0})} \mathbb{E}_M\left[d(T_Q, T_{Q_0})\right]$ is the Wasserstein distance that measures the minimum transportation cost required to transform $\mathcal{T}_{Q_0}$ to $\mathcal{T}_Q$, and $\pi(\mathcal{T}_Q, \mathcal{T}_{Q_0})$ denotes all joint distributions for $(\mathcal{T}_Q, \mathcal{T}_{Q_0})$ with marginals $\mathcal{T}_Q$ and $\mathcal{T}_{Q_0}$. AQP aims to find the most challenging query distribution $\mathcal{T}_Q$ for an original query distribution $\mathcal{T}_{Q_0}$ that lies within or on the Wasserstein ball $B_\rho(\mathcal{T}_{Q_0})$. The hardest perturbation to the query distribution $\mathcal{T}_{Q_0}$ is the one that maximizes the model's query loss $L^*$. Updating the model using such difficult query distribution $\mathcal{T}_Q$ improves its generalizability. Further, the transformation of $\mathcal{T}_{Q_0}$ into $\mathcal{T}_Q$ induces a distributional disparity in a new virtual task comprising of the original support set from $\mathcal{T}_{S_0}$ and the projected query set from $\mathcal{T}_Q$. A model adapted to such virtual tasks is compelled to extract the shift-invariant representations from $T_{S_0} \sim \mathcal{T}_{S_0}$ transferable to $T_Q \sim \mathcal{T}_Q$ to reduce the query loss $L^*$. As adversarial perturbations are adaptive to the model's state, they do not have a monotonic structure throughout the meta-training phase. The evolving augmentations expose the model to diverse SQS. A model meta-trained on such virtual tasks with different SQ shifts learns to extract diverse shift-invariant representations increasing the model's endurance to unknown meta-test SQS. The simultaneous restrain of $\mathcal{T}_Q$ to a Wasserstein ball radius $\rho$ ensures $\mathcal{T}_Q$ does not deviate extensively from $\mathcal{T}_{Q_0}$, and $\mathcal{T}_Q, \mathcal{T}_{Q_0}$ share the label space, and $\mathcal{T}_{Q_0}, \mathcal{T}_Q \in H$ is maintained. Thus the newly-framed meta-objective is:

$$\min_{\theta \in \Theta} \sup_{W_d(\mathcal{T}_Q, \mathcal{T}_{Q_0}) \leq \rho} \mathbb{E}_{(T_Q \sim \mathcal{T}_Q)}\left[L^*(\phi, T_Q)\right] \tag{1}$$

where $\phi \leftarrow \theta - \alpha \nabla_\theta L(\theta; T_{S_0})$. Note that ML approaches such as ProtoNet [9] and MatchingNet [14] do not require adaptation, and hence $\theta = \phi$. As equation 1 is intractable for an arbitrary $\rho$, we use Langragian relaxation for a fixed penalty parameter $\gamma \geq 0$ to convert this constrained objective to an unconstrained objective.

$$\min_{\theta \in \Theta} \sup_{\mathcal{T}_Q} \left\{\mathbb{E}_{\mathcal{T}_Q}[L^*(\phi, T_Q)] - \gamma W_d(\mathcal{T}_Q, \mathcal{T}_{Q_0})\right\} \tag{2}$$

This unconstrained objective (equation 2) is strongly concave and hence easy to optimize. It involves maximizing the loss $L^*$ on adversarial query projections $T_Q$ while simultaneously restraining $T_Q$ to a $\rho$ distance from $T_{Q_0}$.

Table 1: Comparison of ML methods with their Ind_OT and AQP counterparts across Cifar 100, miniImagenet, tieredImagenet, FEMNIST datasets, and their SQS and SQS+ variants. The results are obtained on 5-way tasks with 5 support and 8 query instances per class except for FEMNIST and its variants, which contains only one support and one query instance per class. The ± represents the 95% confidence intervals over 2000 tasks. AQP outperforms classic, and Ind_OT-based ML approaches approximately on all datasets.

| Method | Test Accuracy | | | | | |
|---|---|---|---|---|---|---|
| | No SQS | SQS | SQS+ | No SQS | SQS | SQS+ |
| | Cifar 100 | | | miniImagenet | | |
| ProtoNeT | 48.07 ± 0.44 | 43.15 ± 0.48 | 40.59 ± 0.69 | 64.56 ± 0.42 | 41.68 ± 0.76 | 35.17 ± 0.78 |
| Ind_OT+ ProtoNeT | 48.62 ± 0.44 | 43.62 ± 0.49 | 41.74 ± 0.65 | 63.74 ± 0.42 | 39.84 ± 0.78 | 34.75 ± 0.80 |
| AQP+ ProtoNeT | **48.70 ± 0.42** | **45.09 ± 0.46** | **45.06 ± 0.46** | **66.81 ± 0.42** | **42.65 ± 0.57** | **40.61 ±0.60** |
| MatchingNet | 46.03 ± 0.42 | 39.89 ± 0.44 | 36.63 ± 0.45 | 59.68 ± 0.43 | 39.66± 0.54 | 35.40 ±0.52 |
| Ind_OT+ MatchingNet | 45.77 ± 0.42 | 40.82 ± 0.45 | 37.13 ± 0.47 | 59.64 ± 0.44 | 38.25± 0.54 | 33.22± 0.50 |
| AQP+ MatchingNet | **46.53 ± 0.43** | **42.40 ± 0.46** | **41.26 ± 0.46** | **62.29 ± 0.42** | **42.32 ± 0.52** | **37.90 ± 0.53** |
| | tieredImagenet | | | FEMNIST | | |
| ProtoNeT | 71.04 ± 0.45 | 41.59 ± 0.57 | 38.57 ± 0.65 | 93.09 ± 0.51 | 84.36 ± 0.74 | 82.67 ± 0.77 |
| Ind_OT+ ProtoNeT | 69.56 ± 0.46 | 40.08 ± 0.56 | 35.81 ± 0.58 | 91.66 ± 0.55 | 79.64 ± 0.80 | 76.37 ± 0.84 |
| AQP+ ProtoNeT | 69.62 ± 0.45 | **45.34 ± 0.60** | **40.94 ± 0.66** | **94.61 ± 0.45** | **85.92 ± 0.69** | **84.42 ± 0.74** |
| MatchingNet | 67.85 ± 0.46 | 43.30 ± 0.56 | 37.57 ± 0.57 | 93.69 ± 0.49 | 85.88 ± 0.69 | 83.48 ± 0.74 |
| Ind_OT+ MatchingNet | 67.79 ± 0.46 | 44.27 ± 0.56 | 39.24 ± 0.59 | **93.76 ± 0.48** | 84.08 ± 0.71 | 83.09 ± 0.74 |
| AQP+ MatchingNet | **68.40 ± 0.45** | **45.26 ± 0.56** | **39.39 ± 0.58** | 93.69 +- 0.49 | **87.24 ± 0.67** | **84.98 ± 0.72** |

**3.2.1 Estimation of AQP.** We employ gradient ascent with early stopping on the query set instances $X^*$ to find their corresponding adversarial query projections $X_w^*$. Specifically, we perform an iterative gradient ascent on $X^*$ using $L^*$, resulting in an augmented query set $X_w^*$. This augmented query set $X_w^*$ has distributional disparity with original support set $X$. Early stopping regularizes $(-\gamma d(T_Q, T_{Q_0}))$ and ensures $X_w^*$ does not deviate extensively from $X^*$.

## 4 Experiments and Results

We design experiments to investigate the challenging nature of our proposed SQS+ benchmark and empirically validate the efficacy of the proposed AQP over the state-of-the-art approach to address SQS in inductive settings. We consider Cifar 100, miniImagenet, tieredImagenet, FEMNIST, and their state-of-the-art SQS variants for evaluation. We also demonstrate the AQP's efficiency on our proposed SQS+ versions of benchmark datasets. The SQS+ versions of Cifar 100, miniImagenet, and tieredImagenet datasets are constructed from their SQS counterparts [3] by removing perturbations from the meta-train datasets. Similarly, the SQS+ variant of FEMNIST also follows its SQS counterpart, but the meta-train set contains alpha-numerals from users randomly. We add these SQS+ versions of benchmark datasets to the FewShiftBed [3]. We used Conv4 models [3] for Cifar 100, FEMNIST and their variants, and ResNet-18 [9] for miniImagenet, tieredImagenet, and their extensions. We use $32 \times 32$ images for Cifar 100, $28 \times 28$ for FEMNIST, and $84 \times 84$ for miniImagenet and tieredImagenet. The modified FewShiftBed, which includes the proposed solution, details of SQS+ versions of datasets, and implementation details, is publicly available.[1]

### 4.1 Evaluation of SQS+

We first validate that SQS+ is more challenging than the SQS problem [3]. We train Prototypical and Matching networks on Cifar 100, miniImagenet, tieredImagenet, and FEMNIST on all three settings - No SQS, SQS, and SQS+. We report the results in Table 1 and observe that for all the datasets, models trained with both the approaches (ProtoNet and MatchingNet) perform best in the No SQS

---

[1] `https://github.com/Few-Shot-SQS/adversarial-query-projection`

setting, followed by SQS and SQS+. In the classical few-shot setting, meta-train and meta-test phases share the domain, due to which the meta-knowledge is easily transferable across the phases. However, in SQS, each task's support and query set represent different domains, but share a latent structure, during the meta-train and meta-test phases. In SQS versions of Cifar 100, miniImagenet, and tieredImagenet, both meta-train and meta-test SQS are characterized by different types of data perturbations. However, in FEMNIST's SQS variant, meta-train and meta-test SQS is induced due to different writers. A meta-model trained in this setup becomes partially resilient to the related but disjoint SQS during meta-testing. A common SQS structure across meta-train and meta-test sets may not exist. Thus, SQS+ datasets are more challenging, which is empirically validated by the baseline approach's poor performance.

## 4.2 Evaluation of AQP

We compare the efficiency of the proposed AQP and OT based state-of-the-art solution in handling vanilla SQS and SQS+ on the benchmark datasets. A strong baseline for SQS+ is the inductive version of OT (Ind_OT), where we employ OT only in the meta-train phase to generate projected support sets using support and query instances of a task. We evaluate ProtoNet and MatchingNet versions of Ind_OT and AQP. Table 1 presents the results for this evaluation. We observe that the models learned on projected support data obtained by Ind_OT are less robust to both SQS and SQS+ than the models learned on AQP for all approaches and datasets. Hence, AQP is better at addressing SQS+ (and SQS), when meta-test unlabeled query instances are unavailable. To inspect whether the proposed AQP negatively impacts the models' generalization in the absence of meta-test SQS, we evaluate the ML approaches and their Ind_OT and AQP counterparts on classic datasets containing no support query shifts (No SQS). We observe from Table 1 that AQP does not lead to degradation in the performance in the absence of SQS, instead improves the generalizability of the model even when SQS is absent. The use of different architectures across the datasets shows the robustness of a model trained via AQP across architectures.

## 5 Conclusion and Future Directions

This paper proposes SQS+ - a more challenging distribution shift between the support and query sets of a task in a few-shot meta-learning setup. SQS+ includes an unknown SQ shift in the meta-test tasks, and empirical evidence suggests SQS+ is a complex problem than the prevalent SQS notion. We propose Adversarial Query Projection (AQP) to address SQS+ without leveraging unlabelled meta-test query instances. Exhaustive experiments involving AQP on multiple benchmark datasets (Cifar 100, miniImagenet, tieredImagenet, and FEMNIST - their SQS and proposed SQS+ variants), different architectures, and ML approaches demonstrate its effectiveness. We incorporate proposed AQP and SQS+ versions of Cifar 100, miniImagenet, tieredImagenet, and FEMNIST to FewShiftBed and make it publicly available to encourage research in this direction. The future work includes verifying the effectiveness of AQP in complex SQ shifts, e.g., shift from real to sketch images and creating datasets corresponding to these difficult SQ shifts.

## 6 Limitations and Broader Impact Statement

We evaluated AQP in the cases where the perturbations in data characterize SQS, and for FEMNIST dataset, different writers characterize SQS. More complex SQ shifts may exist in real-world problems - drastic changes may occur in data acquisition from support to query, or a shift from sketch images in support outlined by a domain expert to real query pictures may exist. AQP's performance is not verified for these cases yet. Nevertheless, AQP is a baseline for addressing SQS+, and the publically available resources will help the ML community. We declare that our work has no ethical implications and contains no human subject experiments.

**Acknowledgements**. The support and the resources provided by 'PARAM Shivay Facility' under the National Supercomputing Mission, Government of India at the Indian Institute of Technology, Varanasi and under Google Tensorflow Research award are gratefully acknowledged.

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
