# OpenReview forum: "Adversarial Projections to Tackle Support-Query Shifts in Few-Shot Meta-Learning"
_automl.cc/AutoML/2022/Workshop/Late_Breaking — AutoML 2022 (Late-Breaking Workshop)_

### Meta-Review · Area_Chair_xtgd · 2022-05-09

**Recommendation:** Accept
**Confidence:** 4

**Metareview:**

This paper proposes to study a problem where there may be distribution shifts within each task at test time, between its support and query sets. They also propose an approach for this that is less transductive compared to previous work (doesn’t assume access to unlabeled query data during meta-testing), which is based on perturbing query sets during meta-training. The reviewers found that the paper is clearly motivated, the empirical evidence shows the usefulness of the proposed method for robustness to support-query shift at test time, while it remains competitive with other methods when no such shift is present. The reviewers also found the paper is clearly written for the most part, though I recommend the authors to incorporate reviewers’ feedback about clarity and missing technical details when revising the paper. It would also be interesting to further discuss how this relates to shifts in meta-training vs meta-testing distribution in terms of difficulty and relevance to different real-world applications (though the two may be orthogonal). Would the proposed model work well on a harder benchmark where there are meta-train / meta-test shifts (e.g. as in cross-dataset few-shot learning) in addition to SQ shifts?

---

### Decision · Program_Chairs · 2022-05-13

Accept